# miR-152-3p Represses the Proliferation of the Thymic Epithelial Cells by Targeting *Smad2*

**DOI:** 10.3390/genes13040576

**Published:** 2022-03-24

**Authors:** Ying Li, Xintong Wang, Qingru Wu, Fenfen Liu, Lin Yang, Bishuang Gong, Kaizhao Zhang, Yongjiang Ma, Yugu Li

**Affiliations:** College of Veterinary Medicine, South China Agricultural University, Guangzhou 510642, China; liyingzp93@163.com (Y.L.); wxt614@126.com (X.W.); 18238793929@163.com (Q.W.); lf15802025308@163.com (F.L.); yl19990201666@163.com (L.Y.); gongbishuang@hotmail.com (B.G.); zkz12320092006@163.com (K.Z.); mayongjiang@scau.edu.cn (Y.M.)

**Keywords:** miR-152-3p, *Smad2*, thymic epithelial cells, cell proliferation

## Abstract

MicroRNAs (miRNAs) control the proliferation of thymic epithelial cells (TECs) for thymic involution. Previous studies have shown that expression levels of miR-152-3p were significantly increased in the thymus and TECs during the involution of the mouse thymus. However, the possible function and potential molecular mechanism of miR-152-3p remains unclear. This study identified that the overexpression of miR-152-3p can inhibit, while the inhibition of miR-152-3p can promote, the proliferation of murine medullary thymic epithelial cell line 1 (MTEC1) cells. Moreover, miR-152-3p expression was quantitatively analyzed to negatively regulate *Smad2*, and the *Smad2* gene was found to be a direct target of miR-152-3p, using the luciferase reporter assay. Importantly, silencing *Smad2* was found to block the G1 phase of cells and inhibit the cell cycle, which was consistent with the overexpression of miR-152-3p. Furthermore, co-transfection studies of siRNA–*Smad2* (si*Smad2*) and the miR-152-3p mimic further established that miR-152-3p inhibited the proliferation of MTEC1 cells by targeting *Smad2* and reducing the expression of *Smad2*. Taken together, this study proved miR-152-3p to be an important molecule that regulates the proliferation of TECs and therefore provides a new reference for delaying thymus involution and thymus regeneration.

## 1. Introduction

The thymus is an important immune organ both in animals and humans. It mainly comprises the thymocytes and thymic stromal cells at different stages of maturity. The thymic epithelial cells (TECs) constitute the thymic stromal cells, providing a specific microenvironment for differentiating, developing, and maturing thymocytes [1]. However, the thymus gradually begins to degenerate with aging after the organism reaches sexual maturity. In-depth studies of thymus involution revealed differences in the size and structure of the thymus at different ages. The number of thymocytes and TECs were found to mainly decrease [2,3]. The thymus of the male mouse has been reported to reach its maximum at the age of 1 month (1 M), and thymocyte populations were found to decrease sharply from 1 to 3 months of age (3 M) until this trend of decrease attained stability [4,5]. Studies have shown the destruction of the normal thymic microenvironment to mainly reduce the numbers of TECs [5,6], and the process of thymus involution in mice was found to be delayed or slowed down by promoting the proliferation of TECs [7,8,9]. Therefore, TECs were found to be vital for thymic involution, and changes in TEC number were found to affect age-related thymic involution [10].

MicroRNAs (miRNAs) are noncoding small RNA molecules that regulate gene expression by directly binding to the 3′ untranslated region (UTR) of target mRNAs [11,12]. Previous studies have demonstrated that miRNAs can participate in biological processes, such as cell differentiation, autophagy, apoptosis, and proliferation [13,14]. In recent years, further studies have suggested that miRNAs may be related to thymic involution [15]. For example, the absence of Dicer or the miR-29a cluster causes rapid loss of thymic cellularity [16]; likewise, miR-181a-5p can regulate TEC proliferation by regulating TGF-β signaling [17], microRNA-195a-5p is known to inhibit the proliferation of TECs by targeting *Smad7* [18], the miR-205-5p-mediated FA2H-TFAP2A feedback loop inhibits the proliferation of TECs and participates in the age-related involution of the mouse thymus [19], and miR-199b-5p promotes the proliferation of TECs by activating Wnt signaling by targeting *Fzd6* [20]. In our previous study, the expression of miRNAs was compared in mouse thymus tissues and TECs of different ages using miRNA microarray and RNA-seq technology. When the differential expression profiles of the thymus and TECs were analyzed in mice, miR-152-3p expression levels were found to increase significantly with age [21,22,23,24] (Appendix A). However, the mechanism by which miR-152-3p is involved in thymic involution has not yet been elucidated.

*Smad* belongs to the SMAD family of proteins, divided into the receptor-regulated SMADs (R-SMADs), common-mediator SMADs (co-SMADs), and inhibitory SMADs (I-SMADs). *Smad2* and *Smad3* belong to the R-SMADs [25,26,27]. SMAD2 is a downstream protein involved in the transforming growth factor signaling pathway which regulates multiple signals, such as proliferation, differentiation, and apoptosis [28,29,30,31]. At present, there are very few reports on *Smad2* affecting TECs and regulating thymic involution. This study investigated the effect of miR-152-3p on proliferating murine medullary thymic epithelial cell line 1 (MTEC1) cells and studied the regulation of gene expression between miR-152-3p and *Smad2* and its role in regulating thymic involution.

## 2. Materials and Methods

### 2.1. Animals

This study involved 20 male SPF BALB/c mice, ages 1 M and 3 M, with 10 mice of each age. All the animal experiments in this study were conducted according to the protocol approved by the Animal Protection Committee of South China Agricultural University (Guangzhou, China).

### 2.2. Isolation of the Thymus Tissues and TECs

miR-152-3p expression was detected and verified from the isolated thymic tissues (5 mice per age) and TECs (5 mice per age) under sterile conditions. The thymus tissues were isolated from the mice in each age group with sterile forceps and stored in the tissue tube in liquid nitrogen (−196 °C). The method of isolating and purifying the TECs was performed according to a previous description [19,32]. Briefly, the chopped thymus was digested in collagenase for 30 min at 37 °C. Then, the termination solution (Dulbecco’s Modified Eagle Medium (DMEM) containing 10% fetal bovine serum (FBS)) was added. The collected cell suspension was filtered through an 80 μm cell sieve and then added to the Percoll cell separation solution (Sigma-Aldrich, St. Louis, MO, USA), centrifuged at 600 rcf for 30 min, and the white cell layer was collected. Then, anti-mouse CD45 magnetic beads and anti-mouse CD326 magnetic beads (BD, East Rutherford, NJ, USA) were used to sort and enrich the TECs. Finally, a CytoFLEX flow cytometer (Beckman Coulter Inc., Atlanta, GA, USA) was used to check the purity of the enriched cells.

### 2.3. Culture of the MTEC1 and HEK-293T Cells

The MTEC1 cells were donated by the Peking University Health Science Center (Beijing, China). The MTEC1 cells were cytologically identified as being as densely arranged as the epithelial cells. The immunohistochemical assay confirmed that the cytoplasm of the cells was rich in keratin. Electron microscopy identified the desmosomes between the cells, and the cytoplasm was found to have bundles of tonofilaments, with developed Golgi apparatuses, abundant mitochondria, rough endoplasmic reticula, and high electron-dense granules. This cell line was found to be positive for the rat anti-mouse thymic stromal antibody 33, which corresponded to the antibody 33-positive epithelial cells distributed in the medulla area identified by immunofluorescence staining of the mouse thymus section. Therefore, this cell line was derived from the epithelial cells in the medulla region of the thymus and had functions similar to the murine TECs [33]. The HEK-293T cells were acquired from the American Type Culture Collection (ATCC, Manassas, VA, USA) and all the cells were grown in DMEM supplemented with 10% FBS (Thermo Fisher Scientific, Waltham, MA, USA) in a humidified atmosphere of 37 °C and 5% CO_2_.

### 2.4. RT-qPCR Assays

The total RNA was extracted from the thymus or cells using Trizol reagent (Takara, Kusatsu, Japan), following the protocol. The RNA concentration and purity of each sample were quantified using a NanoDrop ND-2000 (NanoDrop Technologies, Bio Tek, Winooski, VT, USA) according to the following criteria: A260/A280 > 1.8; A260/A230 > 2.0. Then, cDNA was synthesized using the random primers or oligo (DT), following the instructions for the ReverTra Ace qPCR Kit (Toyobo, Osaka, Japan). Using the cDNA as a template, a quantitative reverse transcription polymerase chain reaction (RT-qPCR) was performed using SYBR Green real-time PCR Master Mix (Toyobo, Osaka, Japan). The reaction conditions were 5 min at 95 °C, followed by 40 amplification cycles of 10 s at 95 °C and 30 s at 60 °C. β-actin and U6 were used as the internal controls of mRNA and miRNA, respectively. The primer software Premier 5.0 (PREMIER Biosoft International, Palo Alto, CA, USA) was used to design the mRNA primers (Table 1), while the miRNA primers were designed according to instructions by RiboBio Co. (Guangzhou, China). Finally, the expression level was determined by the 2^−^^ΔΔ^^CT^ method.

### 2.5. Cell Viability Assay

The MTEC1 cells were inoculated in a 96-well plate at a density of 5 × 10^3^ cells/well and cultured at 37 °C in an incubator containing 5% CO_2_ for 12 h. Then, 50 nM miR-152-3p mimic, 100 nM miR-152-3p inhibitor, or mimic/inhibitor negative control (mimic/inhibitor-nc) (RiboBio, Guangzhou, China) were transfected into the MTEC1 cells using Lipofectamine 3000 (Invitrogen, Carlsbad, CA, USA), according to the manufacturer’s instructions. After 48 h of transfection, 10 µL CCK-8 (MedChemExpress, Monmouth, NJ, USA) was added to each well and reacted in an incubator containing 5% CO_2_ at 37 °C for 2 h. Subsequently, the absorbance of the MTEC1 cells plated in a 96-well plate was measured at OD_450_ nm using an automatic microplate reader (Thermo Fisher Scientific), and the influence of miR-152-3p expression on the viability of MTEC1 cells was analyzed.

### 2.6. Cell Proliferation Assay

The MTEC1 cells were inoculated into a 12-well plate at a confluence of 1 × 10^5^ cells per well and were transfected with miR-152-3p mimic, miR-152-3p inhibitor, and negative control according to the method mentioned above. After 48 h of culture, 500 μL of medium containing 50 μmol/L 5-ethynyl-2′-deoxyuridine (EdU) was added to each well. Subsequently, the transfected MTEC1 cells were fixed and stained according to the steps outlined for the EdU kit (RiboBio, Guangzhou, China). Cell fluorescence was collected and photographed using the Leica DM4000 B LED microscope (Leica Microsystems, Wetzlar, Germany).

### 2.7. Cell Cycle Assay

The MTEC1 cells were transfected with siRNA–*Smad2* (si*Smad2*), miR-152-3p mimic, inhibitor, and negative control, and were, respectively, digested with 0.25% trypsin and collected in a 1.5 mL centrifuge tube. Then, 1 mL of 70% ethanol was added to each tube and fixed at 4 °C for 12 h. The supernatant was discarded after the solution was centrifuged at 1500 rpm for 5 min and the MTEC1 cells were washed twice with PBS. After that, 500 μL of propidium iodide (PI) staining solution was added to the tube and incubated at 37 °C in the dark for 30 min. Finally, the cell suspension was placed on a CytoFLEX flow cytometer (Beckman Coulter Inc., Atlanta, CA, USA) in a 1.5 mL centrifuge tube for detection, and the data were analyzed using ModFit software (Verity Software House, Topsham, ME, USA). This experiment was repeated three times.

### 2.8. Target Genes Prediction

Different prediction tools, including miRBase (https://mirbase.org (accessed on 12 March 2019)), TargetScan (https://www.targetscan.org (accessed on 12 March 2019)), and miRDB (https://mirdb.org (accessed on 12 March 2019)), were used for predicting the target genes of miR-152-3p. Then, the three predictions results were intersected to select the candidate target gene of miR-152-3p. The functions of candidate target genes were predicted according to the Gene Ontology (GO) database (http://geneontology.org (accessed on 18 March 2019)) and the Kyoto Encyclopedia of Genes and Genomes (KEGG) database (http://www.kegg.jp/kegg (accessed on 18 March 2019)). Finally, the candidate target genes related to the cell cycle and cell proliferation were selected for verification.

### 2.9. Dual-Luciferase Reporter Gene Assay

The targeted regulatory relationship between *Smad2* and miR-152-3p was further determined by constructing the reporter gene plasmid of the target gene, *Smad2*. Specifically, the wild-type (WT) fragment of *Smad2* 3′ UTR was amplified using the primers containing the miR-152-3p and *Smad2* binding site (Table 2). Then, the *Smad2* 3′ UTR mutant-type (MUT) fragment was synthesized by mutating 4 bases of the binding site using the Stratagene mutation kit (Stratagene, Heidelberg, Germany). Subsequently, these two fragments were connected to the luciferase reporter vector pmiRGLO (Promega, Madison, WI, USA) and the recombinant plasmids pmiRGLO–*Smad2*-3′ UTR-WT and pmiRGLO–*Smad2*-3′ UTR-MUT were constructed. The extracted recombinant plasmid was then co-transfected with the miR-152-3p mimic into the HEK-293T cells. After 48 h of transfection, the activity of luciferase was detected using a dual-luciferase detection kit (Promega, Madison, WI, USA).

### 2.10. Western Blot Analysis

The Western blot experiments were performed according to a previously published method [20]. Briefly, the total protein of the cells was extracted using RIPA lysis buffer (Sigma-Aldrich, St. Louis, MO, USA), and then the concentration of the protein samples was measured using a BCA kit (Thermo Fisher Scientific, Waltham, MA, USA). The denatured protein was continuously electrophoresed using an SDS-PAGE for 1 h. After transferring the protein from the gel to a PVDF membrane, skimmed milk powder was added and sealed at 25 °C for 1 h. After removing the blocking solution, the primary antibodies (mouse anti-SMAD2, mouse anti-C-MYC, mouse anti-β-ACTIN) were added and incubated overnight at 4 °C. The next day, the primary antibody was recovered, and the secondary antibody was added and incubated for 1 h at 37 °C. Finally, ECL chemiluminescence solution (Solarbio, Beijing, China) was added to the protein membranes away from the light, and the images were collected using the Tannon 5200 chemiluminescence analyzer (Tannon, Shanghai, China).

### 2.11. Statistical Analysis

The statistical software SPSS 17.0 (Chicago, IL, USA) was used for the analyses and the Student’s *t*-test and one-way analysis of variance (ANOVA) were used for comparison between the groups. The mean ± standard deviation (SD) was used to represent the data. A *p* < 0.05 was considered to be statistically significant. Here, *, **, ***, and **** represent *p* < 0.05, *p* < 0.01, *p* < 0.001, and *p* < 0.0001, respectively.

## 3. Results

### 3.1. The Expression of miR-152-3p in the Mice Thymus and TECs

The expression of miR-152-3p in the isolated thymus and TECs from the 1 M and 3 M mice was detected using RT-qPCR. The results indicated the expression of miR-152-3p in the thymus tissues and TECs of the 3 M mice to be significantly higher than in the 1 M mice (Appendix A and Figure 1). This finding was consistent with previous miRNA microarray and RNA-seq data [21,22,23,24], suggesting that miR-152-3p might be crucial for regulating age-related thymic involution.

### 3.2. miR-152-3p Inhibits the Viability and Proliferation of MTEC1 Cells

The function of miR-152-3p was studied using the miR-152-3p mimics and inhibitors for detecting its effects on the MTEC1 cells. The RT-qPCR results indicated a significant change in the expression of miR-152-3p in the MTEC1 cells due to the mimic and inhibitor of miR-152-3p, corroborating that miR-152-3p can be successfully transfected into MTEC1 cells (Figure 2a,b). CCK-8 detection revealed the viability of the MTEC1 cells to be significantly down-regulated when transfected with 50 nM of the miR-152-3p mimic, while the administration of 100 nM of the miR-152-3p inhibitor was found to increase the viability of the MTEC1 cells (Figure 2c). Furthermore, the EdU results demonstrated the numbers of fluorescently stained cells in the miR-152-3p mimic group to be lower than in the mimic-nc group (Figure 2d). Conversely, the numbers of fluorescent cells in the miR-152-3p inhibitor group were found to increase compared to the inhibitor-nc group (Figure 2e) (Appendix A). These data confirmed that miR-152-3p overexpression can significantly inhibit MTEC1 cell proliferation.

### 3.3. miR-152-3p Represses the Cell Cycle of MTEC1 Cells

To explore the inhibitory effect of miR-152-3p on the proliferation of MTEC1 cells, the effect of miR-152-3p on the cell cycle was analyzed. After 48 h of transfection, the number of MTEC1 cells in the G1 phase of the miR-152-3p mimic group increased sharply compared to the mimic-nc group, and MTEC1 cell numbers were found to significantly decrease in the S phase. However, compared with the inhibitor-nc group, the miR-152-3p inhibitor group was found to have fewer MTEC1 cells in the G1 phase, and the cells in the S phase were found to increase (Figure 3a). The expression levels of cell cycle-related genes were further detected, as shown in Figure 3b, and the gene expression levels of *C-myc*, *Ccnd1*, *Ccne1*, and *Cdk4* in the miR-152-3p mimic group were found to significantly decrease relative to those in the mimic-nc group. When the expression of miR-152-3p in the MTEC1 cells was inhibited, the gene expression levels of *C-myc*, *Ccnd1*, *Ccne1*, and *Cdk4* were found to significantly increase (Figure 3c). These results suggest that miR-152-3p overexpression leads to G1 phase arrest and cell cycle inhibition, thereby inhibiting MTEC1 cell proliferation.

### 3.4. Putative Target Genes of miR-152-3p

For identifying the mRNAs targeted by miR-152-3p, the GO and KEGG databases were analyzed for 344 predicted target genes from the intersection of different prediction websites, and 10 candidate genes (*Ywhab*, *Tgif2*, *Ltbp1*, *Fbn1*, *Acvr1*, *Mdm4*, *Smad2*, *Cdc14a*, *Dnmt1*, and *Tgfb2*) related to the cell cycle and cell proliferation were screened. Then, 10 candidate gene expression levels were detected in the MTEC1 cells transfected with the miR-152-3p mimic, mimic-nc, miR-152-3p inhibitor, and inhibitor-nc. *Smad2* expression was significantly down-regulated in the miR-152-3p mimic group, while *Ltbp1*, *Fbn1*, and *Acvr1* expressions were significantly up-regulated (Figure 4a). In contrast, *Smad2* expression was significantly up-regulated in the miR-152-3p inhibitor group (Figure 4b). The Western blot results showed that the SMAD2 expression corresponded to the results of the RT-qPCR (Figure 4c and Appendix A), collectively indicating that *Smad2* may be a target gene of miR-152-3p.

### 3.5. Validation of the Target Gene Smad2

The miRNA target tool was used to determine the relationship between *Smad2* and miR-152-3p, and this miRNA was found to directly bind to the 745–751 bp (UGCACUG) and 6263–6269 bp (UGCACUG) sites of *Smad2* 3′ UTR. As shown in Figure 5a, the two recombinant reporter plasmids of *Smad2* containing WT and MUT, respectively, were designed. The miR-152-3p mimic and recombinant reporter plasmid were co-transfected into HEK-293T cells, and the luciferase activity of the pmirGLO–*Smad2*-3′ UTR-WT + miR-152-3p mimic group was found to be significantly lower than that of the pmirGLO–*Smad2*-3′ UTR-WT + mimic-nc group. However, the co-transfection group of pmirGLO–*Smad2*-3′ UTR-MUT + miR-152-3p mimic (Figure 5b) showed no significant change. These results showed that miR-152-3p could bind to the seed sequence of *Smad2* 3′ UTR, and *Smad2* was the direct target gene regulated by miR-152-3p in the MTEC1 cells.

### 3.6. miR-152-3p Regulated the Proliferation of MTEC1 Cells by Targeting Smad2

To explore the biological functions of *Smad2*, si*Smad2* were transfected into the MTEC1 cells. The quantitative results revealed that the expression of *Smad2* was suppressed, and the expression of *C-myc*, *Ccnd1*, *Ccne1*, and *Cdk4*, related to the cell cycle, was found to be significantly reduced (Figure 6a). In Figure 6b, the SMAD2 protein in the si*Smad2* group decreased compared to the control (Appendix A). The flow cytometry study verified that after transfection of si*Smad2*, cell number in the G1 phase increased significantly, and the cellular population in the S phase and G2 phase decreased significantly (Figure 6c). These results were consistent with the results of miR-152-3p overexpression. Subsequently, si*Smad2* and miR-152-3p mimics/inhibitors were co-transfected into the MTEC1 cells. The mRNA expression of *Smad2*, *C-myc*, *Ccnd1*, *Ccne1*, and *Cdk4* in the si*Smad2* + mimic group was found to be significantly reduced compared to the siNC + mimic-nc group. A similar trend was observed in the si*Smad2* + inhibitor group (Figure 6d). These results indicated that miR-152-3p affects cell cycle progression by targeting *Smad2*, thereby regulating the proliferation of MTEC1 cells.

## 4. Discussion

MicroRNAs critically regulate a series of developmental and physiological pathways, including cell proliferation, differentiation, and apoptosis. In addition, miRNAs take part in constructing the thymus microenvironment, affecting the development, maturation, and involution of the thymus [15,34,35]. When the differential expression profiles of the thymus and TECs were analyzed in the 1 M and 3 M mice, miR-152-3p was found to be highly expressed in the 3M group. In our study, we confirmed that miR-152-3p expression was elevated in age-related thymic involution. miR-152-3p is reportedly related to the proliferation of a variety of cells. For instance, miR-152-3p can promote cell proliferation and invasion in the keloid fibroblasts by targeting *Foxf1* [36]. Furthermore, miR-152-3p has been identified in multiple studies as being involved in regulating the occurrence and development of cancer. For example, Yin et al. [37] proved that miR-152-3p can target *Cdk8* to inhibit the occurrence of liver cancer. In 2020, Zhu et al. [38] reported that miR-152-3p inhibition or overexpression of KLF4 could reduce the expression of *Ifitm3* in colon cancer cells, weakening the ability of the cancer cells to proliferate. However, so far, no studies have reported the regulation of proliferation of TECs by miR-152-3p. This study used mouse MTEC1 cells and showed that over-expression of miR-152-3p in MTEC1 cells could significantly reduce cell viability, blocking the G1 phase and inhibiting cell proliferation, while inhibiting miR-152-3p expression had the opposite effect. These data indicated that miR-152-3p could regulate the cell cycle and affect the proliferation of MTEC1 cells. To further study its mechanism, *Smad2* was predicted to be the target gene of miR-152-3p based on the negative regulation characteristics of miRNA on the target genes.

The SMAD protein has three domains: the N-terminal Mad Homology 1, C-terminal Mad Homology 2, and the linking region. The N-terminal domain recognizes the specific DNA sequences, while the C-terminal domain accounts for the protein–protein interactions [39]. The SMAD2 protein links comprise multiple phosphorylation sites that can be phosphorylated by the different kinases to execute different functions and participate in regulating the TGF-signaling pathways. For example, when the *Smad2*/*3*–SMAD4 complex translocates into the nucleus, the phosphorylation site of *Smad2*/*3* is phosphorylated by CDK8 and CDK9, combining *Smad2*/*3* with the other transcription cofactors to improve transcriptional activity [40]. On the other hand, the *Smad2*/*3* phosphorylation sites can be phosphorylated by GSK3, enabling the E3 ubiquitin ligase NeDD4-L to recognize *Smad2*/*3*, promoting its degradation and inhibiting the TGF-β signaling pathway [41]. SMAD2 is a downstream protein in the transforming growth factor signaling pathway, responsible for regulating multiple signals, such as proliferation, apoptosis, and differentiation. Studies have found that LNC-LFAR1 expression inhibition can inhibit bile duct cancer cell proliferation, migration, and invasion, and reduce the expression of *Vimentin*, *Tgf-β1*, *Smad2*, and *Smad4* [42]. Furthermore, the increase in *Tgf-β1* and decrease in *Smad2* in the non-T cells of the mouse thymus after burn injury suggested that TECs might account for increased thymic apoptosis [43]. However, it has not been reported whether miR-152-3p regulates cell proliferation by targeting *Smad2* in TECs. In our study, the dual-luciferase experiment demonstrated that miR-152-3p is targeted to bind *Smad2*, inhibiting its expression. Furthermore, up-regulation of miR-152-3p in MTEC1 cells could induce G1 arrest by targeting *Smad2*. Conversely, down-regulation of miR-152-3p could promote the proliferation and cell cycle progression of MTEC1 cells. Our results verified that the proliferation of TECs is negatively regulated by miR-152-3p via the targeting of *Smad2*. In addition, miR-152-3p overexpression or suppression of *Smad2* was also confirmed to inhibit the expression of *C-myc*, triggering G1 phase arrest. This corroborates previous reports stating that *C-myc* participates in cell proliferation by regulating G1-related genes and protein kinases [44]. *Smad2* acts upstream of *C-myc* in the cell cycle pathway; hence, *Smad2* inhibition can reduce the expression of *C-myc*. Therefore, miR-152-3p can be speculated to indirectly regulate *C-myc* in inhibiting MTEC1 cellular proliferation. However, the detailed mechanism of miR-152-3p in regulating the cell cycle and cell proliferation remains to be further investigated.

## 5. Conclusions

Our study proved that miR-152-3p is an important regulatory molecule in thymic involution. miR-152-3p can inhibit the proliferation of TECs cells by targeting the 3′ UTR of *Smad2*. The cell cycle signaling pathway regulated by *Smad2* is crucial for miR-152-3p to regulate the proliferation of TECs. Although there are no further studies to identify other direct targets of miR-152-3p, our experiments preliminarily suggest that up-regulation of miR-152-3p can be used as a potential biomarker in age-related thymic involution.

## Figures and Tables

**Figure 1 genes-13-00576-f001:**
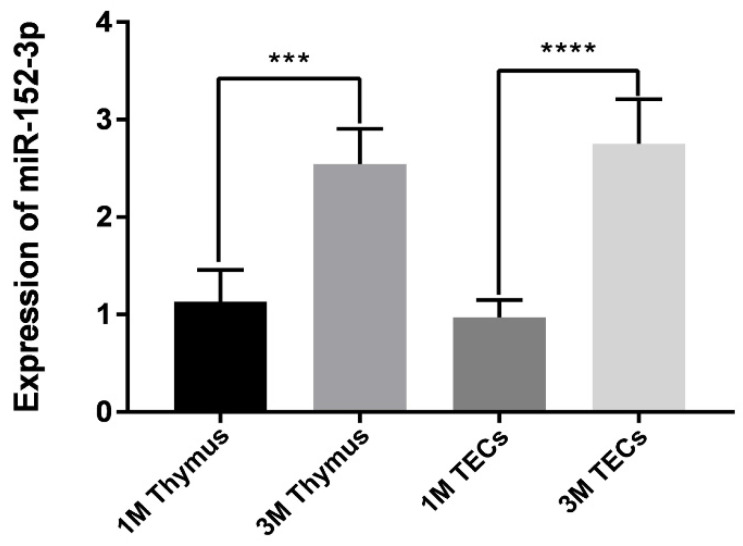
The expression of miR-152-3p from isolated thymuses and thymic epithelial cells (TECs). The miR-152-3p from the thymuses and TECs of 1 M and 3 M mice were quantified using RT-qPCR. Data represent the mean ± standard deviation (SD) of three biological replicates (Student’s *t*-test: ***, *p* < 0.001; ****, *p* < 0.0001).

**Figure 2 genes-13-00576-f002:**
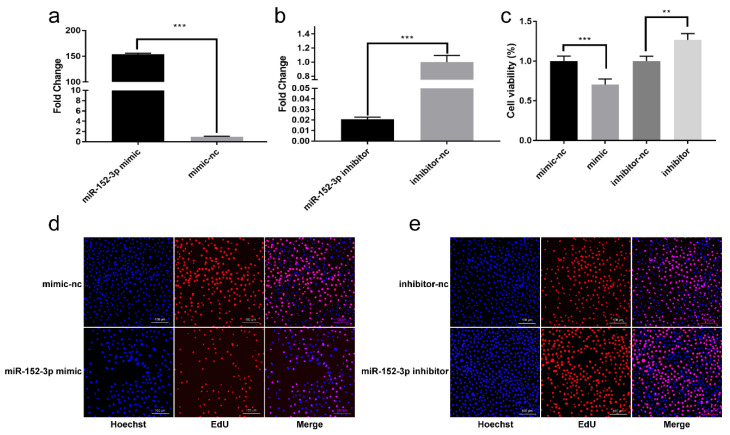
The effect of miR-152-3p on the viability and proliferation of the murine medullary thymic epithelial cell line 1 (MTEC1) cells. (**a**,**b**) The transfection efficiency of the miR-152-3p mimic and inhibitor in the MTEC1 cells. (**c**) CCK-8 assays to examine the proliferation of MTEC1 cells. (**d**,**e**) The proliferation effect of the MTEC1 cells was detected using the 5-ethynyl-2′-deoxyuridine (EdU) kit. While Hoechst stains the nucleus blue, EdU reacts with Apollo stain to turn the replicating cells red. Magnification: 100×. The RT-qPCR data were normalized to the level of U6. Data were expressed as mean ± SD for three biological replicates (Student’s *t*-test: **, *p* < 0.01; ***, *p* < 0.001).

**Figure 3 genes-13-00576-f003:**
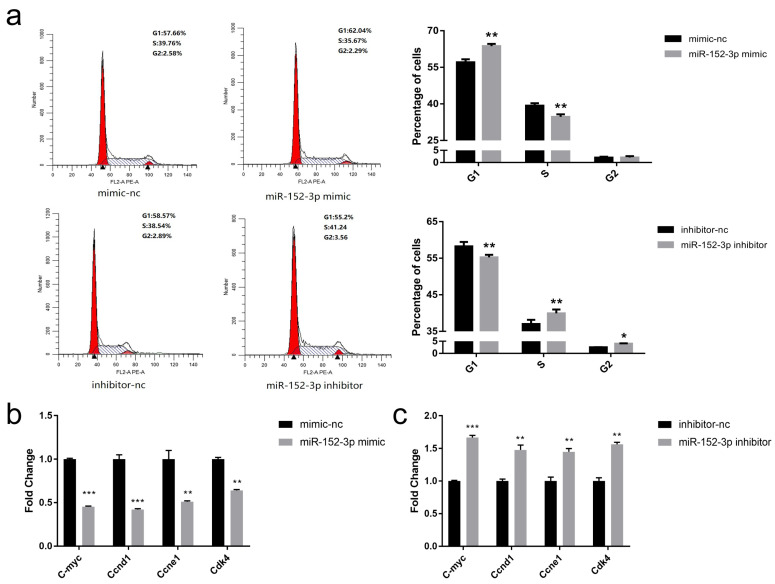
The effects of miR-152-3p on the cell cycle of MTEC1 cells. (**a**) The cell cycle of MTEC1 cells was detected by flow cytometry when the miR-152-3p was overexpressed and inhibited. (**b**,**c**) RT-qPCR was used to detect the expression of the cell cycle-related genes in the MTEC1 cells. The RT-qPCR data were normalized to the level of β-actin for mRNA. Data were expressed as mean ± SD for three biological replicates (Student’s *t*-test: *, *p* < 0.05, **, *p* < 0.01; ***, *p* < 0.001).

**Figure 4 genes-13-00576-f004:**
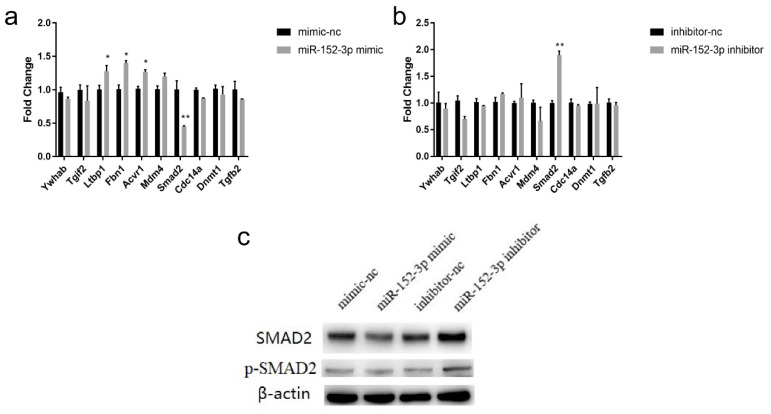
Screening of the miR-152-3p target genes. (**a**,**b**) The effect of miR-152-3p gain and silencing in the MTEC1 cells on the expression of 10 candidate target genes. (**c**) SMAD2 protein expression level in the MTEC1 cells. The RT-qPCR data were normalized to the level of β-actin for mRNA. Data were expressed as mean ± SD for three biological replicates (Student’s *t*-test: *, *p* < 0.05; **, *p* < 0.01).

**Figure 5 genes-13-00576-f005:**
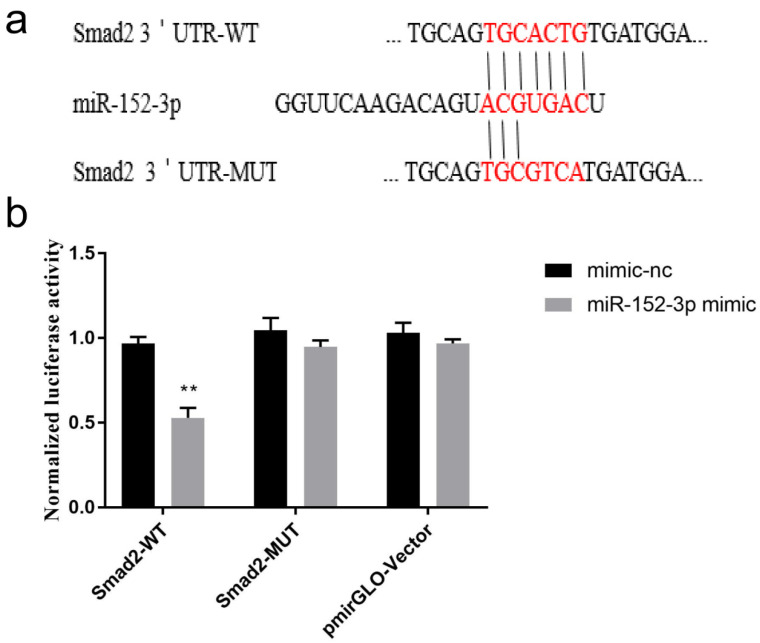
The direct target gene of miR-152-3p is *Smad2.* (**a**) The binding sites and mutant bases of miR-152-3p and *Smad2* 3′ UTR. (**b**) Luciferase activity of the recombinant plasmids pmirGLO–*Smad2*-3′ UTR-WT or pmirGLO–*Smad2*-3′ UTR-MUT co-transfection with the miR-152-3p mimic. The RT-qPCR data were normalized to the level of U6. Data were expressed as mean ± SD for three biological replicates (Student’s *t*-test: **, *p* < 0.01).

**Figure 6 genes-13-00576-f006:**
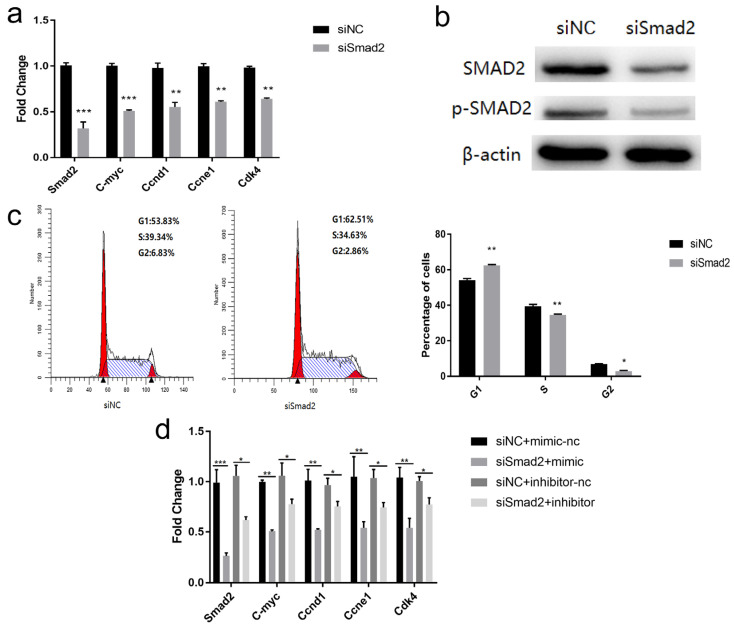
miR-152-3p inhibits the proliferation of MTEC1 cells by targeting *Smad2*. (**a**) The expression level of the cell cycle-related genes was detected in the MTEC1 cells transfected with siRNA–*Smad2* (si*Smad2*). (**b**) si*Smad2* decreased the expression of the SMAD2 protein in the MTEC1 cells. (**c**) The effect of si*Smad2* on MTEC1 cell cycle progression was measured using flow cytometry. (**d**) After co-transfection with miR-152-3p and si*Smad2*, cell cycle-related gene expression was detected in the MTEC1 cells. The RT-qPCR data were normalized to the level of β-actin for mRNA. Data were expressed as mean ± SD for three biological replicates (Student’s *t*-test: *, *p* < 0.05; **, *p* < 0.01; ***, *p* < 0.001).

**Table 1 genes-13-00576-t001:** Primer sequences for the quantitative reverse transcription polymerase chain reaction (RT-qPCR).

Genes	Forward and Reverse Primer Sequences (5′–3′)	Product Size (bp)
*C-myc*	F: GCAACTTCTCCACCGCCGAT	
NM_010849.4	R: AACCGCTCCACATACAGTCCT	141
*Ccnd1*	F: CATGAACTACCTGGACCGCTT	
NM_007631.2	R: TGCTTGTTCTCATCCGCCTCT	285
*Ccne1*	F: GTGCGAAGTCTATAAGCTCCA	
NM_007633.2	R: CGCCATCTGTAACATAAGCAA	194
*Cdk4*	F: CTACATACGCAACACCCG	
NM_009870	R: TCAAAGATTTTCCCCAACT	118
*Ywhab*	F: CCGGAGAAAATAAACAAACCAC	
NM_018753.6	R: CAATCGCCTCATCAAATGCC	201
*Tgif2*	F: TCTGCACCGCTACAACGCCTA	
NM_173396.3	R: ACTGATTAGGGTCTTTGCCAT	158
*Ltbp1*	F: GCTCTTTCCGCTGCCTCTGTTATC	
NM_019919.4	R: AGTTCACACTCGTTCACATCCACAC	82
*Fbn1*	F: GAGTGCCAAGAAATCCCGAAC	
NM_007993.2	R: AATCGTGTTTCTGCAAGTCCC	174
*Acvr1*	F: ATCGCTTCAGACATGACCTCC	
NM_001110204.1	R: TCCGAAGGCAGCTAACCGTA	127
*Mdm4*	F: TTGTTTCAGACACTACGGATGA	
NM_001302801.1	R: GTTTGCTCAGAATTAGCAGCTT	100
*Smad2*	F: CTCTCCAACGTTAACCGAAATG	
NM_010754.5	R: CACCTATGTAATACAAGCGCAC	82
*Cdc14a*	F: CTTACAACCTCACCGTCCT	
NM_001080818.2	R: TATTCTTCCGCATCAAACGTCT	94
*Dnmt1*	F: GAGACGAAAAACGACACGTAAA	
NM_001199432.1	R: CACTTTGGTGAGTTGATCTTCG	117
*Tgfb2*	F: CTGTACCTTCGTGCCGTCT	
NM_009367.4	R: GCCATCAATACCTGCAAATCTCG	82
*β-actin*	F: CATCCGTAAAGACCTCTATGCCAA	
NM_007393.5	R: ATGGAGCCACCGATCCACA	171

**Table 2 genes-13-00576-t002:** Primer sequences of the *Smad2* 3′ untranslated region (UTR).

Target Gene	Primer Sequence (5′–3′)	Accession No.	Product Length
*Smad2* 3′UTR	F: GA*GCTC*TCTTGTAACAGAAACCGTGTG R: GT*CGAC*AATAGTGTCCACCTTCCGAG	NM_010754.5	671 bp

Note: The italics indicate restriction sites (*Sac I* and *Sal I*).

## Data Availability

Raw data on the cell cycle of this study are available in Appendix A. Other data reported in this study are available from the corresponding author upon reasonable request.

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
