# Peer review of "miR-152-3p Represses the Proliferation of the Thymic Epithelial Cells by Targeting Smad2"

_genes, 2022, doi:10.3390/genes13040576_

Round 1

Reviewer 1 Report

In the present study, the authors investigated the effect of miR-152-3p on the proliferation of thymic epithelial cells and proposed SMAD2 as a downstream target of this microRNA using both in vitro and in vivo approaches. My specific comments are listed below:

  • Lines 83-85: What was the percentage of viability? Have you validated the bead separation using flow cytometry? 
  • Lines 102-104: What was the purity of RNA?
  • Lines 105-107: Please describe the PCR condition used. 
  • Table 1: The sequence for internal control is missing. Please also list the amplicon length for each of the genes. 
  • Lines 193-195: The citation is missing.
  • Figure 2c: Please change the Y-axis label to 'Cell viability (%)'.
  • Figure 3a: How many times the experiment was repeated? Please share the raw data with statistical analysis. 
  • Figure 6c: How many times the experiment was repeated? Please share the raw data with statistical analysis. 

Author Response

请参阅附件。

Reviewer 2 Report

In the manuscript titled ‘miR-152-3p represses the proliferation of the thymic epithelial cells by targeting Smad2’, the authors hypothesized that miR-152-3p is associated with thymic involution as its expression significantly increases in thymus and TECs of aged mice. To test this hypothesis, the authors performed various gain- and loss-of-function experiments in vitro. They found that overexpression of miR-152-3p in the thymic epithelial cell line 1 (MTEC1) induces G1 phase arrest and inhibits cell proliferation by negatively regulating Smad2, a mediator of the TGF-β pathway, directly. This study highlights the potential role of miR-152-3p as a biomarker for thymic involution. However, here are a few concerns regarding the manuscript-

  1. In lines 193-194 authors mention that ‘This finding was consistent with the previous data on the miRNA microarray and RNA-seq, suggesting that miR-152-3p might be crucial for regulating age-related thymic involution’. It is important to quote these studies and authors can incorporate the gene expression data from RNA-seq and microarray in the manuscript to further strengthen this point.
  2. The authors show that the cell proliferation activity of MTEC1 reduces upon transfection with miR-152-3p mimic. Since the number of Hoechst-stained cells is less in Fig. 2d bottom panel than mimic-nc, the authors should check if reduced proliferation is caused by increased apoptosis.
  3. The authors should provide quantification of cell proliferation assays represented in Fig. 2d and 2e. Scale bars should be inserted in the microscopy images. Authors should also account for the intense blue background in Hoechst-stained images or put better representative images.
  4. It is unclear what the authors are denoting as cell activity (%) on the y-axis of Fig 2c. If the graph represents the readout of the CCK-8 assay as stated in lines 204-205, the y-axis should be cell viability (%). The figure legend should also be corrected accordingly.
  5. The differences between nc control vs mimic or inhibitor in Fig. 3 are very minor. In fact, the FACS plots don’t show any difference between nc vs mimic. Y-axis between nc and inhibitor in Fig. 3b are different and that also contributes to minor visual differences in G1 population across these plots. Similarly, the expression c-myc and other genes in Fig. 3d is induced to only ~1.5 fold upon miR-152-3p inhibitor treatment. Moreover lack of information on statistics in figure legend makes it more complicated. It is difficult to interpret if these minor differences will contribute to the biological phenomenon.
  6. The authors should include quantification of Smad2 levels shown in Fig. 4c in the manuscript.
  7. In Fig. 6d, the comparison should be made between siNC+mimic vs siSMAD2+mimic. Similarly between siNC+inhibitor vs siSMAD2+inhibitor. siNC+mimic and siNC+inhibitor data are missing. The idea is to demonstrate that silencing of SMAD2 and mimic have additive effect. The * added on bars don’t really tell which two datasets are being compared.
  8. The authors should mention the statistical tests performed and the number of biological replicates in figure legends.
  9. Authors don’t have conclusive data to suggest that miR-152-3p is involved in mTEC proliferation. Rather the data may suggest that this miRNA is involved in the maintenance of mTECs. The title of the article should be modified accordingly.
